# Invasive vs. Invasive, *Parthenium hysterophorus* as a Bio-Control Agent against Invasive Mealybugs

**Taswar Ahsan** [1] , **Bingxue Li** [1,*] **and Yuanhua Wu** [2,*]

1 College of Land and Environment, Shenyang Agricultural University, Shenyang 110866, China; taswarahsan@163.com
2 College of Plant Protection, Shenyang Agricultural University, Shenyang 110866, China
* Correspondence: libingxue@syau.edu.cn (B.L.); wuyh09@syau.edu.cn (Y.W.); Tel.: +86-139-981521288 (Y.W.)

**Abstract:** *Parthenium hysterophorus* has an antagonistic potential against mealybugs, which are hosted on wild *Dalbergia sissoo*. In the current study, an aqueous extract of *Parthenium* was evaluated against mealybugs on *Dalbergia sissoo*. A serial dose of 12.5, 25, 50, 100, 200, and 500 μg/mL of aqueous extract was prepared from all parts of the *Parthenium* plant. After 72 h at high doses, 200 μg/mL and 500 μg/mL aqueous extracts had high mortality of 76.67% and 73.33% via the residual method, respectively. Meanwhile at same dose after 72 h, the contact method had higher mortality percentages of 80% and 80% at 200 μg/mL and 500 μg/mL, respectively. After 48 h at a high dose (200 μg/mL and 500 μg/mL), the mortality of the mealybugs was highest via the contact method. The results show a mortality of 73.33% with both doses. Meanwhile the same doses, via the contact method, after 48 h had a 63.33% mortality rate. After 24 h via the contact method, at 200 μg/mL and 500 μg/mL the mortality of mealybug was 70% with both doses, whereas via the mortality rate via the residual method at 200 μg/mL and 500 μg/mL doses was 56.67% and 66.67%, respectively. These results indicate that *Parthenium* is a strong bio-control agent against mealybugs. Aqueous extracts could lead to a cost effective and environmentally friendly insecticidal for sustainable use in large scale forestry.

**Keywords:** *Parthenium hysterophorus*; *Dalbergia sissoo*; mealybug





## 1. Introduction

*Dalbergia sissoo* belongs to the *Fabaceae* family, and naturally exists in India, Pakistan, Afghanistan, Iran, Australia, the Americas, and Africa. It is mostly situated near river and canal banks and is currently an endangered species [1]. Recently, an invasive species, the mealybug *Phenacoccus solenopsis Tinsley* (*Hemiptera: Pseudococcidae)* has attacked the cotton and caused heavy economic losses in the Indo-Pakistani region. Mealybugs secrete sticky honeydew, which helps mold grow and results in the disturbance of photosynthesis [2]. It was accidentally spread from America and has caused a sudden outbreak throughout Asia [3]. Mealybugs use the *Dalbergia sissoo* as a secondary host [4].

Pakistan's cotton growing regions (Punjab and Sindh) were badly affected by mealybugs [5]. In 2007, over US$121.4 million was used on pesticides in the Punjab province in only two months aiming at controlling the mealybug outbreak [6]. Under irrigation, the marginally suitable habitats in southern Pakistan, northern India, parts of the Middle East, and some Uygur regions in China are also suitable for *P. solenopsis* colonization and establishment [3]. There are several chemicals available on the market, but they are hazardous towards biodiversity and the environment. Biological control is an environmentally friendly approach. There is an urgent need to screen out novel, cost effective, environmentally friendly, and reliable bio-control agents [7]. Plant extracts are have significant insecticidal potential [8], and ecologically based pest management practices have reduced the use of chemicals [9].

*Parthenium hysterophorus* is a toxic weed, which could have negative effects on biodiversity and the environment. However, it could be used as a bio-control agent, such

as an insecticidal, antimicrobial agent, and anticancer agent, as it contains allelopathic compounds and several vital minerals [10]. However, Parthenium is harmful for both plants and animals. However, the whole plant or its parts may have several applications in pharmacy, agriculture, and industry [11]. In the current study, *Parthenium hysterophorus* is used as a botanical biocontrol agent to control mealybugs on *Dalbergia sissoo*. Both are invasive species in Asia. A cost effective approach has been investigated to control mealybug infestation. *Dalbergia sissoo* is already on the red list, so there is an urgent need to control this bug, as *Dalbergia sissoo* is a secondary host of mealybug infestation. According to our investigation, this is the first report to use the *Parthenium hysterophorus* against the mealybugs on *Dalbergia sissoo.*

## 2. Material and Methods

### 2.1. Collection of Plants and Mealybugs

*Parthenium hysterophorus* was collected from the meadow in the Gujarat District of Pakistan with latitudinal and longitudinal gradients (32°34′16.1184′′ N, 74°4′30.0180′′ E), in April 2020. The plant was authenticated at the Department of Botany, University of Gujrat, Pakistan. Mealybugs were collected from the wild *Dalbergia sissoo* tree, from the surrounding area of the University of Gujrat, Pakistan. During the collection of Mealybugs, it was ensured that no pesticides were applied to the plants. The population of the bugs was maintained on *Dalbergia sissoo* plants in a preserved area under natural conditions. The average annual temperature is 24.0 °C/75.1 °F in Gujrat. Precipitation here is about 706 mm | 27.8 inches per year.

### 2.2. Aqueous Extract of Parthenium hysterophorus

An aqueous extract of *Parthenium hysterophorus* was prepared in distilled water. A total of 300 g of mixed parts of the plant were ground and mixed with 1000 mL water. The debris was removed by filter paper, centrifuged to obtain an aqueous suspension and stored at −4 °C for further study. The centrifuge machine Lab-o-check (made in Italy) was used, at 4 °C and a speed of 2000 r/min for 5 min.

### 2.3. In Vitro Assay, Residual Toxicity Methods

The study of the aqueous extract of *Parthenium hysterophorus* against the mealybug was evaluated in vitro. For the in vitro assay, residual toxicity methods were adopted. Serial concentrations such as 12.5 µg/mL, 25 µg/mL, 50 µg/mL, 100 µg/mL, 200 µg/mL, and 500 µg/mL were prepared and replicated thrice. Fresh *Dalbergia sissoo* leaves were cut off and dipped for 10 s in respective concentrations and dried in air for half an hour. Next, 10 adult mealybugs without sex determination were released on these leaves contained in plastic Petri dishes. For control (Ck), Chlopyrifos was prepared in water. All the Petri dishes were placed in an incubator at 20 °C + 5 °C with 65 RH for three days, along with a 16/8 h light/dark cycle [12]. An illustration of this method is given in Figure 1.

### 2.4. In-Vitro Assay, Contact Methods

The study of the aqueous extract of *Parthenium hysterophorus* against the mealybug was evaluated in vitro. For the in vitro assay, direct contact methods were adopted. Serial concentrations such as 12.5 µg/mL, 25 µg/mL, 50 µg/mL, 100 µg/mL, 200 µg/mL, and 500 µg/mL were prepared and replicated thrice. Ten adult mealybugs without sex determination were dipped for 5 s in the respective doses and then left on fresh *Dalbergia sissoo* leaves. For the control (Ck), Chlopyrifos was prepared in water. All the Petri dishes were placed in an incubator at 20 °C + 5 °C with 65 RH for three days along with along with a 16/8 h light/dark cycle [12].

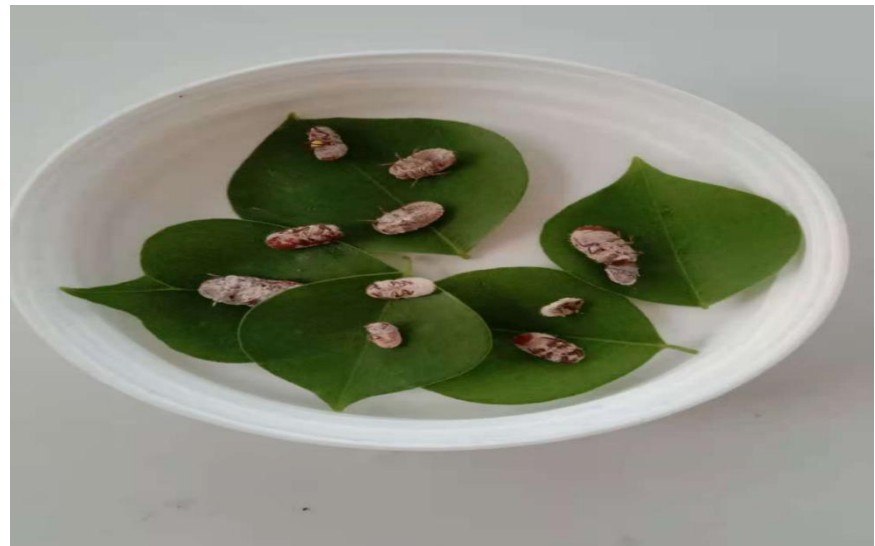

**Figure 1.** Illustration of bio-assay of mealybugs by aqueous extract of *Parthenium hysterophorus*.

*2.5. Data Acquisition and Data Process*

After 24, 48, and 72 h, data on mortality were calculated. The insects were probed with needles, and if there was no motion they were considered dead. An analysis of variance (ANOVA) test was performed to calculate the mortality. The mean mortality variation among the treatment was calculated at $p = 0.05$ by the Duncan multiple range test. Data Processing System version 7.05 Hangzhou Ruifeng information technology co. LTD, (Hangzhou, China) was used.

**3. Results and Discussion**

In the current study, the mortality of the mealybugs shows that parthenium has a strong insecticidal efficacy. As earlier reported, parthenium had insecticidal activity [13]. In our study, we used an aqueous extract of parthenium. As previously reported, aqueous extracts of parthenium had allelopathic potential [14]. The mortality data, presented in Table 1, shows the residual efficacy of aqueous extracts of parthenium against the mealybugs. The mortality rate over time, methods, and concentration interaction show strong insecticidal effects. After 72 h at high doses, 200 μg/mL and 500 μg/mL, aqueous extracts had high mortality rates of 76.67% and 73.33% via the residual method, respectively, whereas with the same dose after 72 h the contact method had a higher mortality rate of 80% and 80% at 200 μg/mL and 500 μg/mL, respectively. Likewise, after 48 h at a high dose, such as 200 μg/mL or 500 μg/mL, the mortality of mealybugs was highest via the contact method. The results show that at both 200 μg/mL and 500 μg/mL the mortality was 73.33%. The same doses, meanwhile, via the contact method, after 48 h had a 63.33% mortality rate. After 24 h via the contact method at both 200 μg/mL and 500 μg/mL the mortality of mealybug was 70%, whereas the via residual method at 200 μg/mL and 500 μg/mL doses the mortality rate was 56.67% and 66.67%, respectively. Interaction among the variance is presented as ANOVA. The three-way interaction between method, concentration, and time is less significant, while the two way interaction between time and concentration is more significant (see Table 2). Therefore, the overall result demonstrated that contact method, long time, and high dose predicted strong insecticidal activity. Our results are supported by the study of [12].

**Table 1.** Mortality of mealybugs on *Dalbergia sissoo* after application of aqueous extract of *Parthenium hysterophorus* via residual and contact method of toxicity.

| Time (h) | Dose (mL) | Method | | Mean |
|---|---|---|---|---|
| | | Residual | Contact | |
| **24 h** | 12.5 | 26.67 ± 3.33 k | 43.33 ± 3.33 jk | 35.00 ± 4.28 G |
| | 25 | 46.67 ± 3.33 ijk | 53.33 ± 3.33 g–j | 50.00 ± 2.58 EFG |
| | 50 | 50.00 ± 5.77 h–k | 60.00 ± 5.77 e–j | 55.00 ± 4.28 DEF |
| | 100 | 63.33 ± 3.33 d–j | 66.67 ± 3.33 c–j | 65.00 ± 2.24 B–E |
| | 200 | 56.67 ± 6.67 f–j | 70.00 ± 5.77 c–i | 63.33 ± 4.94 B–E |
| | 500 | 66.67 ± 3.33 c–j | 70.00 ± 0.00 c–i | 68.33 ± 1.67 BCD |
| | Ck+ | **100.00 ± 0.00 a** | 86.67 ± 3.33 a–d | 93.33 ± 3.33 A |
| **48 h** | 12.5 | 43.33 ± 3.33 jk | 46.67 ± 8.82 ijk | 45.00 ± 4.28 FG |
| | 25 | 60.00 ± 5.77 e–j | 46.67 ± 3.33 ijk | 53.33 ± 4.22 DEF |
| | 50 | 60.00 ± 0.00 e–j | 60.00 ± 0.00 e–j | 60.00 ± 0.00 C–F |
| | 100 | 60.00 ± 5.77 e–j | 70.00 ± 0.00 c–i | 65.00 ± 3.42 B–E |
| | **200** | 63.33 ± 6.67 d–j | **73.33 ± 3.33 b–h** | 68.33 ± 4.01 BCD |
| | **500** | 63.33 ± 3.33 d–j | **73.33 ± 3.33 b–h** | 68.33 ± 3.07 BCD |
| | Ck+ | 96.67 ± 3.33 ab | 90.00 ± 0.00 abc | 93.33 ± 2.11 A |
| **72 h** | 12.5 | 66.67 ± 6.67 c–j | 56.67 ± 8.82 f–j | 61.67 ± 5.43 B–F |
| | 25 | 63.33 ± 3.33 d–j | 63.33 ± 6.67 d–j | 63.33 ± 3.33 B–E |
| | 50 | 70.00 ± 5.77 c–i | 63.33 ± 6.67 d–j | 66.67 ± 4.22 B–E |
| | 100 | 56.67 ± 6.67 f–j | 70.00 ± 5.77 c–i | 63.33 ± 4.94 B–E |
| | **200** | 76.67 ± 3.33 a–g | **80.00 ± 0.00 a–f** | 78.33 ± 1.67 AB |
| | **500** | 73.33 ± 3.33 b–h | **80.00 ± 0.00 a–f** | 76.67 ± 2.11 ABC |
| | Ck+ | 83.33 ± 3.33 a–e | 90.00 ± 5.77 abc | 86.67 ± 3.33 A |

Means sharing similar letter in a row or in a column are statistically non-significant ($p > 0.05$). Small letters represent comparison among interaction means and capital letters are used for overall mean.

**Table 2.** Analysis of variance table for mortality (%).

| Source | Degrees of Freedom | Sum of Squares | Mean Squares | F-Value | Probit |
|---|---|---|---|---|---|
| Method (M) | 1 | 317.5 | 317.46 | 5.00 * | 0.028 |
| Time | 2 | 1961.9 | 980.95 | 15.45 ** | 0 |
| Conc. | 6 | 20,985.7 | 3497.62 | 55.09 ** | 0 |
| M x Time | 2 | 101.6 | 50.79 | 0.80 NS | 0.4527 |
| M x Conc. | 6 | 760.3 | 126.72 | 2.00 NS | 0.0753 |
| Time x Conc. | 12 | 2371.4 | 197.62 | 3.11 ** | 0.0011 |
| M x Time x Conc. | 12 | 1454 | 121.16 | 1.91 * | 0.0446 |
| Error | 84 | 5333.3 | 63.49 | | |
| Total | 125 | 33,285.7 | | | |

M = Method; Conc = concentration; NS = non-significant ($p > 0.05$); * = significant ($p < 0.05$); ** = highly significant ($p < 0.01$).

## 4. Conclusions

The current study reveals that *Parthenium hysterophorus* has an antagonistic potential against mealybug, which is hosted by wild *Dalbergia sissoo*. The contact method had a high mortality rate. No chemicals were used in the main investigation of this study. Therefore, the aqueous extract led to an environmentally friendly and cost-effective approach, especially for large scale forest disease management. This could be an efficient bio-control agent for sustainable forestry. However, there is a need to purify and characterize the pure compound.

**Author Contributions:** T.A. and Y.W. conceived of the idea, T.A. conducted the experiments, B.L. analyzed the data and created the tables, and T.A. and Y.W. wrote the paper. All authors have read and agreed to the published version of the manuscript.

**Funding:** This work was funded by the National Key R&D Program of China (2017YFD0201104); National Key R&D Program of China (2017YFE0104900).

**Data Availability Statement:** The data presented in this study are available on request from the corresponding authors.

**Conflicts of Interest:** The authors declare no conflict of interest.

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
