# Peer review of "Invasive vs. Invasive, Parthenium hysterophorus as a Bio-Control Agent against Invasive Mealybugs"

_forests, doi:10.3390/f12070936_

Round 1
Reviewer 1 Report
General: This paper presents the results of residual and contact tests of an aqueous extract of a major global invasive weed, Parthenium hysterophorus, for insecticidal activity against a major invasive cotton pest in the Indo-Pakistan region as well as in China and Australia. The objective is to develop the extract as a bio-insecticide against the mealybug, and an additional goal is to use the bio-insecticide to protect a rare/endangered plant species, Dalbergia sissoo. The testing protocols appear to be well-established in the laboratory in which the studies were done. I have some concern because the trial was replicated only once for each method (with 3 Petri dishes per dose, for each method). However, the results indicate insecticidal activity that increased over time and as extract concentration increased, as would be expected, but with little or no significant difference between residual and contact exposure methods. The determination of which exposure times and concentrations were successful needs to be made by comparing the percentages to that achieved with the chemical insecticide control. English language review/editing is needed.
Title: Confusing, as it lists the native alternative plant host and then “mealybug”. Please delete plant name and replace with the invasive mealybug name.
Introduction:
L30, L56: Please explain “red list”, as this term is not used worldwide. Indicate the country(ies) in which the plant is on the rare/threatened/endangered list.
Material and Methods:
General: It is good that the studies included a synthetic chemical insecticide positive control.
L78: Indicate centrifugation speed, temperature and length of time, and, if possible, the manufacturer, model number, City and Country of manufacturer.
L85: What stage were the mealybugs in, and how long had they been in that stage prior to exposure? Or indicate if multiple life stages were used. Also, please indicate if male and female mealybugs were added in equal numbers, or indicate that sex could not be determined.
L103: Were the data normally-distributed? If not, were efforts made to transform the data (e.g., arcsine-square root for percentage data) or to run the ANOVA with a non-normal data distribution assumption (e.g., binomial)?
L104: Please provide the full name of the statistical software used, and the manufacturer, City, and Country.
Results and Discussion:
General: The success of the experiments should be evaluated relative to the insecticide control; indicate which treatments and times were statistically the same as that control (at least one common lower-case letter). Those were the successful treatments. Suggest indicating the concentrations and times that were successful in bold font.
Table 2: It does not seem appropriate to average the residual and contact methods results for the final analysis, as they represent different experimental protocols. If other published papers have taken this approach, please cite in your paper, and you can then ignore this comment. You can also refute if you believe that they can be combined because the method x time, and method x concentration interactions were not significant in the ANOVA, and the 3-way interaction was only marginally significant. However, you should state this in the paper.
Conclusion:
L133: Delete “green revolution”, as this is a popular, not scientific term.
L135: Which approach-residual or contact-is more feasible for protection of Dalbergia sissoo against the mealybug? I would suggest residual. Also, which approach is more feasible for protection of cotton against the mealybug? I would suggest contact. If the two exposure methods work together to achieve control, please state this.
General: What level or percentage control of the mealybug do the farmers need to protect cotton from the mealybug, and did the bio-insecticide extract achieve this level of control?
References:
Amoabeng et al 2018, cited on L46, is not listed in References.
Indranil Singh, 2020, cited on L54, is not listed in References.
Is “Masqood et al. 2020”, cited on L99, L123, is the same as Ahmed et al 2020 in References?
Riaz et al. 2015, cited on L109, is not listed in References.
Kaushik Shuchi 2020 is not cited in the text.
Author Response
Uploaded

Reviewer 2 Report
Dear Author, after reviewing your ms with title Invasive vs invasive, Parthenium hysterophorus a bio-control 2 agent against Dalbergia sissoo mealy bug you must check the form of you ms. In the pdf file i have my suggestions

Author Response
Reviewer # 2.
All the changes done, you can find in the main MS, by marking with Track
Changes.
Round 2
Reviewer 2 Report
Dear authors very improved your ms. Please remove fig 1